

# Retrieval and validation of forest background reflectivity from daily MODIS bidirectional reflectance distribution function (BRDF) data across European forests

Jan Pisek[1]*, Angela Erb[2], Lauri Korhonen[3], Tobias Biermann[4], Arnaud Carrara[5], Edoardo Cremonese[6], Matthias Cuntz[7], Silvano Fares[8], Giacomo Gerosa[9], Thomas Grünwald[10], Niklas Hase[11], Michal Heliasz[4], Andreas Ibrom[12], Alexander Knohl[13], Johannes Kobler[14], Bart Kruijt[15], Holger Lange[16], Leena Leppänen[17], Jean-Marc Limousin[18], Francisco Ramon Lopez Serrano[19], Denis Loustau[20], Petr Lukeš[21], Lars Lundin[22], Riccardo Marzuoli[9], Meelis Mölder[4], Leonardo Montagnani[23], Johan Neirynck[24], Matthias Peichl[25], Corinna Rebmann[11], Eva Rubio[19], Margarida Santos-Reis[26], Crystal Schaaf[2], Marius Schmidt[27], Guillaume Simioni[28], Kamel Soudani[29], and Caroline Vincke[30]

[1]Tartu Observatory, University of Tartu, Observatooriumi 1, Tõravere, 61602, Tartumaa, Estonia
[2]University of Massachusetts Boston, Boston, Boston, USA
[3]University of Eastern Finland, Joensuu, Finland
[4]Lund University, Lund, Sweden
[5]Fundacion CEAM, Paterna, Spain
[6]ARPA Valle d'Aosta, Saint Christophe, Italy
[7]Université de Lorraine, AgroParisTech, INRAE, UMR Silva, Nancy, France
[8]CNR-National Research Council, Rome, Italy
[9]Università Cattolica del Sacro Cuore, Brescia, Italy
[10]Technische Universität Dresden, Dresden, Germany
[11]Helmholtz Centre for Environmental Research - UFZ, Leipzig, Germany
[12]Technical University of Denmark, Kongens Lyngby, Denmark
[13]University of Göttingen, Göttingen, Germany
[14]Umweltbundesamt, Vienna, Austria
[15]Wageningen University & Research, Wageningen, Netherlands
[16]Norwegian Institute of Bioeconomy Research, Ås, Norway
[17]Finnish Meteorological Institute, Space and Earth Observation Centre, Sodankylä, Finland
[18]CEFE Univ Montpellier, CNRS, EPHE, IRD, Univ Paul Valéry Montpellier, Montpellier, France
[19]IER-ETSIAM, Universidad de Castilla-La Mancha, Albacete, Spain
[20]INRAe, Bordeaux, France
[21]Global Change Research Institute, Academy of Sciences of the Czech Republic, Brno, Czech Republic
[22]Swedish University of Agricultural Sciences, Uppsala, Sweden
[23]Free University of Bolzano, Bolzano, Italy, and 23b Forest Services, Autonomous Province of Bolzano, Bolzano, Italy
[24]INBO, Geraardsbergen, Belgium
[25]Department of Forest Ecology and Management, Swedish University of Agricultural Sciences, Umeå, Sweden
[26]cE3c – Centre for Ecology, Evolution and Environmental Changes, Lisbon, Portugal
[27]Forschungszentrum Juelich, Juelich, Germany
[28]INRAe - URFM, Avignon, France
[29] Université Paris-Saclay, CNRS, AgroParisTech, Ecologie Systématique et Evolution, 91405, Orsay, France
[30]Université Catholique de Louvain, Louvain-la-Neuve, Belgium

*Correspondence to*: Jan Pisek (janpisek@gmail.com)





**Abstract.** Information about forest background reflectance is needed for accurate biophysical parameter retrieval from forest
canopies (overstory) with remote sensing. Separating under and overstory signals would enable more accurate modeling of
forest carbon and energy fluxes. We retrieved values of normalized difference vegetation index (NDVI) of forest understory
with multi-angular Moderate Resolution Imaging Spectroradiometer (MODIS) bidirectional reflectance distribution function
(BRDF)/albedo data (gridded 500 meter daily Collection 6 product), using a method originally developed for boreal forests.
The forest floor background reflectance estimates from MODIS data were compared with *in situ* understory reflectance
measurements carried out at an extensive set of forest ecosystem experimental sites across Europe. The reflectance estimates
from MODIS data were hence tested across diverse forest conditions and phenological phases during the growing season, to
examine its applicability on ecosystems other than boreal forests. Here we report the method can deliver good retrievals
especially over different forest types with open canopies (low foliage cover). The performance of the method was found limited
over forests with closed canopies (high foliage cover), where the signal from understory gets much attenuated. The spatial
heterogeneity of individual field sites as well as the limitations and documented quality of the MODIS BRDF product are
shown to be important for correct assessment and validation of the retrievals obtained with remote sensing.

## 1 Introduction

The reflectance from the forest canopy background/forest floor can often confound and even dominate the radiometric signal
from the upper forest canopy layer to the atmosphere. Forest understory is defined here as all the components found under the
forest canopy: understory vegetation, leaf litter, moss, lichen, rock, soil, snow, or a mixture thereof (Pisek and Chen, 2009). If
unaccounted for, forest understory can introduce potential bias in the estimation of overstory biophysical parameters (e.g. leaf
area index (LAI), fraction of absorbed photosynthetically active radiation (fAPAR)) and, subsequently, productivity estimates
(e.g. the net primary productivity (NPP)) as the contribution of the understory to the total energy absorption capacity of a forest
stand can be quite significant (Clark et al. 2001; Law et al. 2001). The understory vegetation in forest ecosystems should be
treated differently from overstory in carbon cycle modelling because of different residence times of carbon fixed through net
primary productivity in different ecosystem components (Vogel and Gower, 1998; Rentsch et al., 2003; Marques and Oliveira,
2004; Kim et al., 2016).  Currently, the understory is often treated as an unknown quantity in carbon models due to the
difficulties in measuring it properly and consistently across larger scales (Luyssaert et al., 2007).  The predictions regarding
spectral variation of forest background have posed persistent challenge (McDonald et al., 1998; Gemmell, 2000) because of
the high variability of incoming radiance below the forest canopy, challenges with the spectral characterization and weak
signal in some parts of the spectrum for both overstory and understory (Schaepman et al., 2009), and the general varying nature
of the understory (Miller et al., 1997).

Multi-angle remote sensing can capture signals of different forest layers because the observed proportions for different forest
layers vary with the viewing angle, making it possible to separate forest overstory and understory signal. Here, we aim at
consolidating previous efforts of tracking understory reflectance and its dynamics with multi-angle Earth observation data





(Canisius and Chen, 2007; Pisek and Chen, 2009; Pisek et al., 2010, 2012, 2015a, 2015b, 2016; Jiao et al., 2014) by testing the validity of this approach using Moderate Resolution Imaging Spectroradiometer bidirectional reflectance distribution

function (MODIS BRDF)/albedo data (gridded 500 meter daily Collection 6 MCD43 product) against *in situ* understory reflectance measurements over an extended set of Integrated Carbon Observation System (ICOS) forest ecosystem sites. The validation procedure was defined to comply as much as possible with the best practices proposed by the Committee on Earth Observation Satellites (CEOS) Working Group on Calibration and Validation (WGCV) Land Product Validation (LPV) subgroup (Garrigues et al., 2008; Baret et al., 2006). It corresponds to Stage 1 validation as defined by the CEOS (Nightingale

et al., 2011; Weiss et al., 2014), where product accuracy shall be assessed over a small (typically < 30) set of locations and time periods by comparison with *in situ* or other suitable reference data. Using the extended set of ICOS forest ecosystems as validation sites, we asked the following questions:

1. Can ICOS forest ecosystem sites serve as a suitable validation dataset with respect to their footprint and the pixel resolution

of EO products?

2. Can we retrieve reliable normalized difference vegetation index (NDVI; Rouse et al., 1973) dynamics for understory with MODIS BRDF data across diverse forest conditions during the growing season?

3. Are there important differences between the total (overstory+understory) and understory-only NDVI signal?

## 2 Materials and Methods

### 2.1 Study sites

The Integrated Carbon Observation System (ICOS) is a distributed pan-European research infrastructure providing *in situ* standardized, integrated, long-term and high-precision observations of lower atmosphere greenhouse gas (GHG) concentrations and land- and ocean-atmosphere GHG interactions (Gielen et al., 2017). In this study we carried out the evaluation over the network of 31 ICOS-affiliated forest ecosystem sites, complemented with additional sites in Spain,

Portugal, Austria, and Finland. Together these selected 40 study sites comprise a large variety of forest over- and understory types, spanning a wide latitudinal gradient from almost 38°N (Yeste, Spain) to 68°N (Kenttärova, Finland). Site locations are shown in Figure 1 and vegetation characteristics are summarized in Table 1.

### 2.2 Understory spectra and forest canopy cover/closure in situ measurements

Following the terminology by Schaepman-Strub et al. (2006), we refer to the reflectance factors measured by the field

spectrometers as to the satellite derived hemispherical-directional reflectance factors (HDRFs). The given spectrometer's field of view is approximated as angular (cone) and narrower than a whole hemisphere, with some anisotropy captured which corresponds to normal remote sensing viewing geometry. An overview of the undertaken *in situ* campaigns at each site as well as their characteristics is given in Table 1.





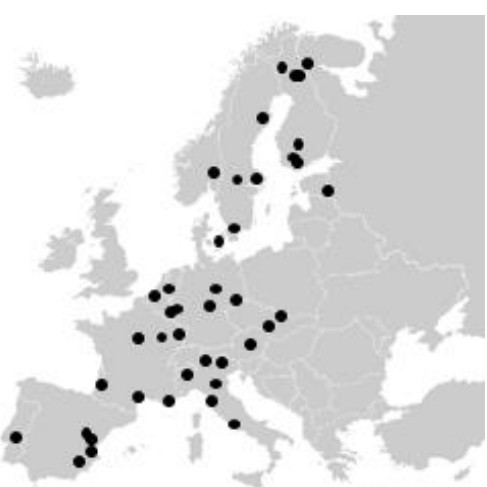


**Figure 1.** Distribution of study sites across Europe; for further details, refer to Table 1.

The individual sites were visited between April 2017 and August 2019, mostly during the growing season. Following the protocol by Rautiainen et al. (2011), the understory spectra were measured with sun completely obscured by the clouds, or

around sunset (diffuse light conditions) covering the visible/NIR region depending on the spectrometer (see Table 1 for more details). Three understory spectra were measured every 2 m along two 50 m long transects laid at each site, resulting in 50 measurement points (150 individual measurements). Transects covered and characterized conditions within the measurement footprint of the given tower. It should be noted that the tower footprint might be different from the exact MODIS pixel footprint (see section 2.5 for the spatial homogeneity assessment of MODIS pixels). The measurements concerned conditions at the

forest floor and low herbaceous and shrubby species or tree seedlings and saplings, as the area sampled by each spectral measurement was estimated to correspond to a ~ 50 cm diameter circle on the ground. The downward-pointed spectroradiometer (no fore-optics were used) was held by the operator's out-stretched hand. Three spectra above a 10-inch Spectralon SRT-99–100 white panel were recorded at the beginning, after every four understory spectra measurement points (every 8 m), and at end of each transect. A hemispherical-conical reflectance factor was obtained with an "uncalibrated"

Spectralon reflectance spectrum and the linearly interpolated irradiance. Finally, broadband HDRFs for red (620–670 nm) and NIR (841–876 nm) wavelengths were computed with relative spectral response functions for the MODIS sensor on-board Terra. Understory NDVI value for given site was calculated from the red and NIR band values and averaged over the two transects.



**Table 1.** Study site characteristics and their spatial representativeness status. ICOS - Integrated Carbon Observing System sites; LTER - Long Term Ecological Research Network sites.

| Site Code | Site Name | Lat (deg) | Lon (deg) | Sampling date | Spectrometer model | Understory vegetation | Representativeness |
|---|---|---|---|---|---|---|---|
| AT_Zbn | Zoebelboden (LTER) | 47.842 | 14.442 | 20160818 | ASD FieldSpec 4 | Calamagrostis varia, Brachypodium sylvaticum, Hordelymus europaeus, Senecio ovatus | Not Representative |
| BE-Bra | Brasschaat (ICOS) | 51.304 | 4.519 | 20190112 | Ocean Optics FLAME-S-VIS-NIR-ES | Betula spec, Quercus robur, Sorbus aucuparia | Representative |
| BE-Vie | Vielsalm (ICOS) | 50.3 | 5.983 | 20180816 | Ocean Optics FLAME-S-VIS-NIR-ES | sparse fern and moss cover | Representative @ 0.5 km |
| CH-Dav | Davos (ICOS) | 46.817 | 9.85 | 20180712 | Ocean Optics FLAME-S-VIS-NIR-ES | dwarf shrubs, blueberry , mosses | Representative |
| CZ-BK1 | Bílý Kříž (ICOS) | 49.502 | 18.539 | 20160417 | ASD FieldSpec 4 | Vaccinium myrtillus L. | Representative @ 1.5 km |
| CZ-Lnz | Lanžhot (ICOS) | 48.682 | 16.948 | 20170427 | ASD FieldSpec 4 | Allium ursinum, Asarum europeum | Representative |
| DE-Hai | Hainich (ICOS) | 51.079 | 10.453 | 20180412 | Ocean Optics FLAME-S-VIS-NIR-ES | Anemone nemorosa, Allium ursinum | Representative |
| DE-HoH | Hohes Holz (ICOS) | 52.083 | 11.217 | 20180411 | Ocean Optics FLAME-S-VIS-NIR-ES | Anemone nemorosa | Representative |
| DE-RuW | Wüstebach (ICOS) | 50.505 | 6.331 | 20180816 | Ocean Optics FLAME-S-VIS-NIR-ES | sparse Deschampsia flexuosa, Deschampsia cespitosa, Molinia caerulea | Not Representative |
| DE-Tha | Tharandt (ICOS) | 50.967 | 13.567 | 20180412 | Ocean Optics FLAME-S-VIS-NIR-ES | Fagus sylvatica, Abies alba, Deschampsia flexuosa | Representative |
| DK-Sor | Soroe (ICOS) | 55.486 | 11.645 | 20180926 | Ocean Optics FLAME-S-VIS-NIR-ES | beech saplings and seedlings, Pteridium Aquilinum | Representative |
| ES-AP1 | Almodovar del Pinar | 39.677 | -1.848 | 20171109 | ASD FieldSpec HandHeld 2 | Quercus ilex ssp ballota, Rosmarinus officinalis, Thymus vulgaris, Lavandula latifolia, Quercus coccifera, Genista scorpius | Representative |
| ES-CMu | Cuenca des Majadas | 40.252 | -1.965 | 20171112 | ASD FieldSpec HandHeld 2 | Juniperus communis, Juniperus oxycedrus, Crataegus monogyna | Representative @ 0.5 km |
| ES-CPa | Cortes de Pallas | 39.224 | -0.903 | 20171108 | Ocean Optics FLAME-S-VIS-NIR-ES | Rosmarinus officinalis, Ulex parviflorus, Brachypodium retusum | Representative > 0.5 km |
| ES-Yst | Yeste | 38.339 | -2.351 | 20180728 | Ocean Optics FLAME-S-VIS-NIR-ES | Rosmarinus officinalis L., Thymus vulgaris L., Cistus clusii Dunal | Representative @ 0.5 km |
| FI-Hal | Halssiaapa | 67.368 | 26.654 | 20170613 | ASD FieldSpec Pro | sedge vegetation | Representative |
| FI-Hyy | Hyytiälä (ICOS) | 61.847 | 24.295 | 20180628 | Ocean Optics FLAME-S-VIS-NIR-ES | Vaccinium spec ., Norway spruce seedlings | Representative @ 0.5 km |
| FI-Ken | Kenttarova (ICOS) | 67.987 | 24.243 | 20170613 | ASD FieldSpec Pro | Vaccinium myrtillus, Empetrum nigrum, Vaccinium vitis-idaea and the forest mosses Pleurozium schreberi, Hylocomium splendens, Dicranum polysetum | Representative @ 2 km |
| FI-Kns | Kalevansuo | 60.647 | 24.356 | 20170615 | ASD FieldSpec Pro | dwarf shrubs, mosses | Representative @ 0.275 km |
| FI-Let | Lettosuo (ICOS) | 60.642 | 23.96 | 20170615 | ASD FieldSpec Pro | dwarf shrubs, mosses, herbs | Representative < 1.0 km |
| FI-Sod | Sodankylä (ICOS) | 67.362 | 26.638 | 20170613 | ASD FieldSpec Pro | lingonberry, Calluna vulgaris, lichens | Spheroid not fit < 1.5 km;Not represenative at > 1.5 km |
| FI-Var | Värriö (ICOS) | 67.757 | 29.616 | 20170614 | ASD FieldSpec Pro | mosses, lichens, dwarf shrubs | Representative |
| FR-Bil | Bilos-Salles (ICOS) | 44.494 | -0.956 | 20180614 | Ocean Optics FLAME-S-VIS-NIR-ES | Molinia coerulea Moench ., Pteridium aquilineum , Ulex europaeus | Representative < 0.5 km |


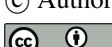



| Site Code | Site Name | Lat (deg) | Lon (deg) | Sampling date | Spectrometer model | Understory vegetation | Representativeness |
|---|---|---|---|---|---|---|---|
| FR-Fon | Fontainebleau-Barbeau (ICOS) | 48.476 | 2.780 | 20180612 | Ocean Optics FLAME-S-VIS-NIR-ES | Carpinus betulus | Representative @ 0.5 km |
| FR-Hes | Hesse (ICOS) | 48.674 | 7.066 | 20180818 | Ocean Optics FLAME-S-VIS-NIR-ES | Fagus sylvatica seedlings, blackberry | Spheroid not fit |
| FR-MsS | Montiers (ICOS) | 48.537 | 5.312 | 20190114 | Ocean Optics FLAME-S-VIS-NIR-ES | sparse Sphagnum spec. vegetation | Representative > 1.0 km |
| FR-Pue | Puechabon (ICOS) | 43.741 | 3.596 | 20180613 | Ocean Optics FLAME-S-VIS-NIR-ES | Buxus sempervirens, Pistacia lentiscus, Phillyrea latifolia, Salvia rosmarinus, Ruscus aculeatus | Representative |
| IT-BFt | Bosco Fontana (ICOS) | 45.202 | 10.743 | 20180710 | Ocean Optics FLAME-S-VIS-NIR-ES | Hedera helix, Corylus spec., Ruscus aculeatus | Representative @ 1.5 km |
| IT-Cp2 | Castelporziano2 (ICOS) | 41.704 | 12.357 | 20190125 | Ocean Optics FLAME-S-VIS-NIR-ES | Phyllirea latifolia, Pistacia lentiscus | Representative @ 1.5 km |
| IT-Ren | Renon (ICOS) | 43.732 | 10.291 | 20180711 | Ocean Optics FLAME-S-VIS-NIR-ES | Deschampsia flexuosa L. , Vaccinium myrtillus L. , Rhododendron ferrugineum L. | Representative @ 0.5 km |
| IT-SR2 | San Rossore (ICOS) | 61.847 | 24.295 | 20180628 | Ocean Optics FLAME-S-VIS-NIR-ES | Ligustrum vulgare | Representative < 1.5 km |
| IT-Trf | Torgnon | 45.833 | 7.567 | 20180707 | Ocean Optics FLAME-S-VIS-NIR-ES | Juniperus communis, Rhododendron ferrugineum, Festuca varia | Not Representative |
| NL-Loo | Loobos (ICOS) | 52.167 | 5.744 | 20180813 | Ocean Optics FLAME-S-VIS-NIR-ES | Prunus serotina, Vaccinium Myrtilus, Deschampsia felexuosa, mosses | Representative @ 0.5 km |
| NO-Hur | Hurdal (ICOS) | 60.372 | 11.078 | 20180927 | Ocean Optics FLAME-S-VIS-NIR-ES | Vaccinium spec., Norway spruce seedlings | Representative |
| PT-Cor | Coruche (LTER) | 39.138 | -8.333 | 20161008 | Ocean Optics FLAME-S-VIS-NIR | Rumex acetosella, Tuberaria guttata, Tolpis barbata Plantago coronopus, Agrostis pourretii, Briza maxima, Vulpia bromoides, V. geniculata | Spheroid not fit |
| SE-Htm | Hyltemossa (ICOS) | 56.098 | 13.419 | 20180928 | Ocean Optics FLAME-S-VIS-NIR-ES | continuous moss cover | Spheroid not fit |
| SE-Knd | Kindla (LTER) | 59.754 | 14.908 | 20160716 | ASD FieldSpec Pro | ericaceous dwarf-shrubs, mosses and lichens | Representative |
| SE-Nor | Norunda (ICOS) | 60.086 | 17.48 | 20181022 | ASD FieldSpec Pro | bilberry, lingonberry, moss | Representative < 1.5 km |
| SE-Svb | Svartberget (ICOS) | 64.256 | 19.775 | 20190823 | ASD FieldSpec Pro | bilberry, lingonberry and moss | Representative |





Estimates of overstory foliage cover and crown cover were obtained from digital cover photographs (DCP). Overstory foliage

cover was defined as the % ground covered by the vertical projection of foliage and branches, and crown cover as the % ground covered by the vertical projections of outermost perimeters of the crowns on the horizontal plane (without double-counting overlap) (Gschwantner et al., 2009). The DCPs were taken from below the canopy every 8 meters along transects at each site. The camera (Nikon CoolPix4500, 2272 x 1704 resolution) was set to automatic exposure, aperture-priority mode, minimum aperture and F2 lens (Macfarlane et al., 2007). The camera was levelled at the height of 1.4 m above the ground and the lens

was pointed towards zenith. This setup provides a view zenith angle from 0 to 15 degrees, which is comparable with the 1st ring of the LAI-2000 instrument (Macfarlane et al., 2007).

We used the algorithm by Nobis and Hunziker (2005) to threshold a majority of the DCP images. However, some of the images were visibly overexposed, i.e. the 8-bit digital numbers (DN) of the background sky were 255 and parts any portion of the sky

black (typically at 240-250 DN). Next, a method based on mathematical image morphology (Korhonen and Heikkinen, 2009) was applied to estimate the foliage and crown cover fractions. In this method, black-and-white canopy images are processed with morphological closing and opening operations that are well known in digital image processing (Gonzalez and Woods, 2002). As a result, a filter for "large" gaps was obtained. When a tuning parameter (called "structuring element" in image processing) was set so that "large" gaps only occurred between individual tree crowns (Korhonen and Heikkinen, 2009), the

proportions of gaps inside and between individual crowns could

### 2.3 Background signal retrieval method with Earth Observation data

The total reflectance of a pixel ($R$) results from the weighted linear combination of reflectance values by the forest canopy, forest background and their sunlit and shaded components (Li and Strahler, 1985; Chen et al., 2000; Bacour and Bréon, 2005; Chopping et al., 2008; Roujean et al., 1992):


$$R = k_T R_T + k_G R_G + k_{ZT} R_{ZT} + k_{ZG} R_{ZG} \quad (1)$$

where $R_T$, $R_G$, $R_{ZT}$, and $R_{ZG}$ are the reflectivities of the sunlit crowns, sunlit understory, shaded crowns, and shaded understory, respectively. $R_G$ marks the BRF of the target (understory). The $k_j$ are the proportions of these components at the chosen view

angle or in the instantaneous field of view of the sensor at given irradiation geometry. Following Canisius and Chen (2007), we derive the understory reflectivity ($R_G$) with the assumption that the reflectivities of overstory and understory at the given illumination geometry differ little between chosen view angles. The most suitable viewing configuration for the retrieval has been identified by Pisek et al. (2015a) using a high angular resolution BRF dataset of Kuusk et al. (2014) and accompanying *in situ* measurements of understory reflectance factors (Kuusk et al., 2013). The configuration consists of the BRF at nadir



**Table 2.** Stand parameters for the Four-Scale model

| Stand | Deciduous | Coniferous |
|---|---|---|
| Stand density (trees/ha) | 500,1000,2000 | 500,1000,2000 |
| Tree height (m) | 25 | 16 |
| Length of live crown (m) | 9.2 | 4.2 |
| Radius of crown projection (m) | 1.87 | 1.5 |
| Leaf area index ($m^2/m^2$) | 1,2,3 | 1,2,3 |

($R_n$ = 0 degrees) with solar zenith angle (SZA) corresponding to the Sun's position at 10:00 local time for given day and another zenith angle ($R_a$ = 40 degrees) with relative azimuth angle PHI = 130 degrees. It can be expressed by the Eqs. (2), (3):

$$R_n = k_{Tn}R_T + k_{Gn}R_G + k_{ZTn}R_{ZT} + k_{ZGn}R_{ZG} \quad (2)$$
$$R_a = k_{Ta}R_T + k_{Ga}R_G + k_{ZTa}R_{ZT} + k_{ZGa}R_{ZG} \quad (3)$$

The proportions of the components ($k_j$) were obtained using the four-scale model (Chen and Leblanc, 1997) with parameters for generalized deciduous and coniferous tree stands as an input (see Table 2) (Kuusk et al., 2013). The understory reflectance at the desired wavelengths can be calculated by combining and solving equations (2) and (3) and insertion of $R_n$ and $R_a$ estimates derived from appropriate Earth Observation data,. The individual components (sunlit/shaded overstory and understory) cannot be resolved with the MODIS spatial resolution. The reflectances of shaded tree crowns ($R_{ZT}$) and understory ($R_{ZG}$) are related to sunlit ones via M as $R_{ZT}$ = M· $R_T$ and $R_{ZG}$ = M· $R_G$, where M = $R_Z/R$ for a reference target, which can be measured in the field, or predetermined with the four-scale model. Here, the same M is assumed for overstory trees as well as understory. Based on his field work in Canadian boreal forests, White (1999) suggested that an angularly constant, wavelength dependent M values may be appropriate, at least during the growing season. The input stand parameters from Table 2 may not be always precisely known while retrieving the understory signal over larger areas. Figure 2 shows the relationships between the available *in situ* data for tree heights or tree densities over our study sites with the 1-km$^2$ resolution estimates from the global maps of Simard et al. (2011) and Crowther et al. (2015). The weak relationships indicate the current unsuitability of the site-specific variable estimates of interest (tree height, tree density) from currently available global maps at a given spatial resolution for our purpose. At the same time, the calculated mean values for the tree heights of needleleaved (17.5 m) and broadleaved tree stands (22.7 m) from Simard et al. (2011) over the study sites were reasonably close to our original generalized input parameter values in Table 2. Following Gemmell (2000), we opted to report a range of understory NDVI (NDVIu) values obtained with the combinations of parameter values from Table 2 for each site and date. Specifying the correct constraints (window) for background alone has been previously found to greatly reduce the errors in the estimation of overstory parameters (Gemmell, 2000).


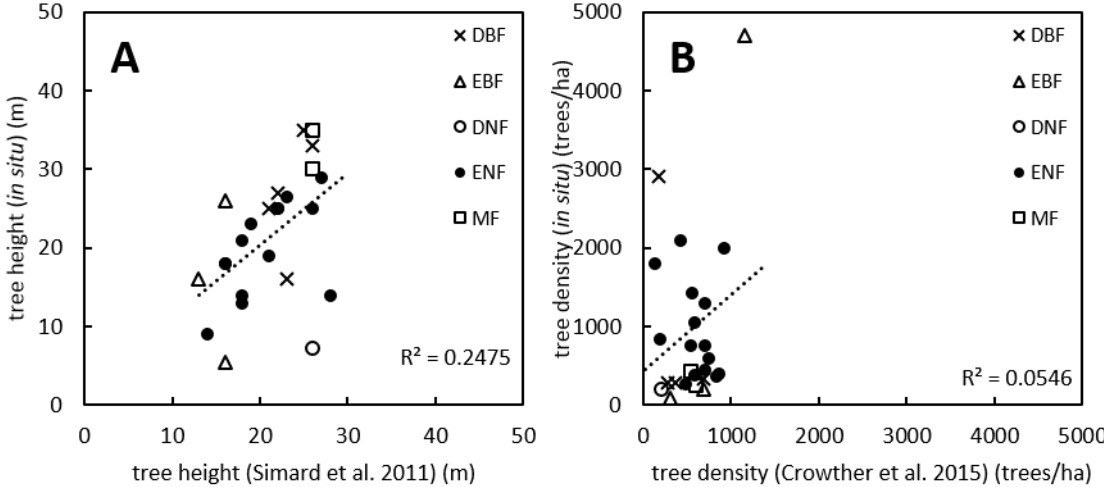

**Figure 2.** (A) Relationship between available *in situ* estimates of tree height (m) with Simard et al.'s, (2011) estimate; (B) relationship between available *in situ* estimates of tree density (trees/ha) with Crowther et al.'s (2015) estimates (DBF-deciduous broadleaf forest; EBF – evergreen broadleaf forest; DNF – deciduous needleleaf forest; ENF – evergreen needleleaf forest; MF – mixed forest).

## 2.4 MODIS BRDF data

The MCD43A1 V6 Bidirectional Reflectance Distribution Function and Albedo (BRDF/Albedo) model parameter dataset is a 500 meter gridded daily product. MCD43A1 is generated by inverting multi-date, multi-angular, cloud-free, atmospherically corrected, surface reflectance observations acquired by MODIS instruments on board the Terra and Aqua satellites over a 16-day period (Wang et al., 2018). The Julian date represents the 9th day of the 16-day retrieval period, and consequently the observations are further weighted to estimate the BRDF/Albedo for that particular day of interest. The MCD43A1 algorithm uses all high quality observations that adequately sample the viewing hemisphere to fit an appropriate semiempirical BRDF model (the RossThickLiSparse-reciprocal model, Roujean et al., 1992; Lucht et al., 2000) for that location and date of interest. We obtained the understory signal with the isotropic parameter and two (volumetric and geometric) kernel functions (Roujean et al., 1992) for MODIS band 1 (red, 620–670nm) and band 2 (NIR, 841–876nm). We used them to reconstruct the bidirectional reflectance factor (BRF) values for required geometries (see section 2.3) for each date. The associated data quality (MCD43A2) product was employed to assess the effect of retrieval quality on the accuracy of the calculated understory signal. All MODIS data have been accessed and processed through Google Earth Engine (Gorelick et al., 2017).

## 2.5 Spatial representativeness assessment of the validation sites

A method developed by Román et al. (2009), and refined by Wang et al. (2012, 2014, 2017) was adopted for evaluating the spatial representativeness of *in situ* measurements to assess the uncertainties arising from a direct comparison between field-



measured forest understory spectra and the corresponding estimates with MODIS BRDF data. To characterize the spatial representativeness of a test site to represent a satellite retrieval, this method uses three variogram model parameters (the range,

sill, and nugget), obtained by the analysis of near nadir surface reflectances from cloud free 30 m Landsat/Operational Land Imager (OLI) data (Román et al., 2009) collected as close to the sampling date as possible. Where valid imagery was not available within a reasonable window of the sampling date, imagery from the corresponding season of a different year was used. As such, the analysis was done to illustrate the representativeness of the tower site with respect to a particular point in time.


Campagnolo et al. (2016) showed that the effective spatial resolution of 500 m gridded MODIS BRDF product at mid latitudes is around 833 m by 618 m because of the varied footprint of the source multi-angular surface reflectance observations. We analyzed each site with five different spatial extents (0.275 km, 0.5 km, 1 km, 1.5 km, 2 km) to assess and illustrate the changes in spatial representativeness with different spatial resolutions.

**3 Results and Discussion**

**3.1 Spatial representativeness**

Table 1 provides the assessment of spatial heterogeneity for all sites included in this study using OLI subsets acquired around the time of *in situ* measurements. The example results for the ICOS sites Norunda (SE-Nor) in Sweden and Wüstebach (DE-RuW) in Germany, using three OLI subsets, are shown in Figure 3. The variogram functions with relevant model parameters

for the two sites are displayed in Figure 3B and D. The range corresponds to the value on the x-axis where the model flattens out. There is no further correlation of a biophysical property associated with that point beyond the range value. The sill is the ordinate value of the range. Smaller sill value indicates a more homogenous surface (less variation in surface reflectance). Surface can be considered spatially representative with respect to the MODIS footprint when the sill value is < 5.0e-04 (Román et al., 2009; Wang et al., 2017). The sill values for all spatial extents are well below the value of 5.0e-04 up to 1 km spatial

resolution in case of Norunda (Figure 3B), which indicates that the field measurements shall be representative and allow comparison with MODIS retrievals at a 500 m spatial resolution. While the Wüstebach site can be considered spatially homogeneous within the immediate vicinity of 275 m around the tower, the sill value exceeds the criteria of 5.0e-04 at > 0.5km spatial resolution. During late summer/early autumn of 2013, trees were almost completely removed in an area of 9 ha west of the tower in order to promote the natural regeneration of near-natural deciduous forest from spruce monoculture forest. The

clear-felling area can be seen on Figure 3D. This action resulted in increasing the spatial heterogeneity


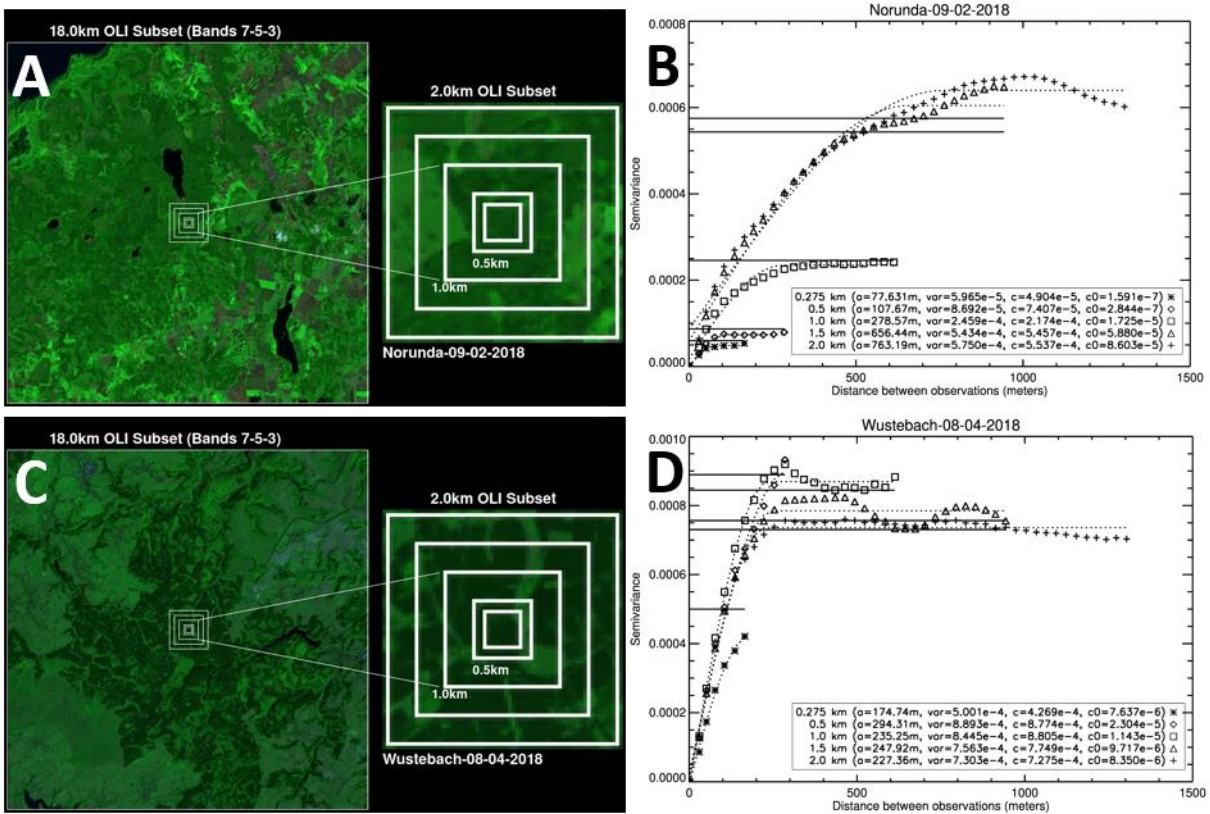

**Figure 3.** Shortwave BRF composites centered at ICOS sites (A) Norunda in Sweden and (C) Wüstebach in Germany. (B,D) Variogram estimators (points), spherical model results (dotted curves), and sample variances (solid straight lines) obtained over the sites with OLI subsets and spatial elements of 0.275 km, 0.5 km, 1.0 km, 1.5 km, and 2 km as a function of distance between observations. Variogram legend explanations: a - variogram range; var - sample variance; c - variogram sill; c0 = nugget variance.

of this ICOS site. *In situ* measurements collected within the footprint of the Wüstebach tower thus cannot be deemed fully comparable with the retrievals with MODIS at a 500 m spatial resolution. Overall most of the sites were found representative at the spatial resolution of MODIS BRDF gridded data. The non-representative cases and the effect on the understory signal retrieval and agreement with corresponding *in situ* measurements carried within the measurement footprint of the individual towers are further discussed in Sections 3.2 and 3.3. Román et al. (2009) provide further details on the assessment of spatial representativeness using a set of four geostatistical attributes derived from semivariograms.

## 3.2 NDVI ranges

There is only a weak relationship between the total (overstory + understory) NDVI signal retrieved with MODIS BRDF data and corresponding *in situ* understory NDVI measurements ($R^2=0.19$; Figure 4). Total NDVI values alone do not allow one to disentangle the correct understory signal. In contrast, our retrieval method could track understory signal dynamics over a





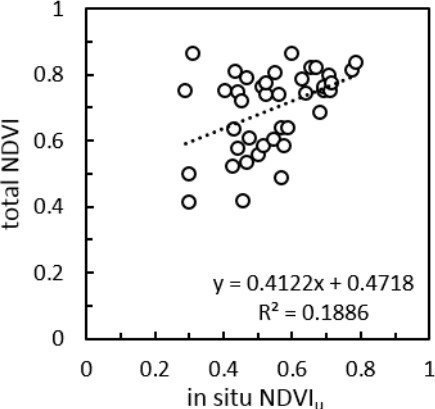

**Figure 4.** Relationship between total (overstory+understory) NDVI values computed from nadir NDVI values using MODIS BRDF/albedo
data and *in situ* measured understory NDVI values over the study sites.

broad NDVI range (Figure 5). The predicted understory NDVI ranges were beyond the uncertainty limits of *in situ* understory
measurements (corresponding to ± 1 SD here) in less than 15% of cases. These sites with poor retrievals were carefully
investigated to identify the issues precluding good results. Below we focus on a discussion of results where the predicted and
*in situ* measured NDVI ranges of understory layer did not agree.

The understory dominated the overall signal of open shrubland at the Cortes de Pallas (ES-CPa) site and the deciduous
broadleaf forest site at Montiers (FR-MsS) during the leaf-off part of the season (Figure 5). Both sites were found to be spatially
representative for comparison with MODIS footprint data at the time of available *in situ* measurements (Table 1). There are
only a very few trees scattered across the Cortes de Pallas site, and ground vegetation is fully exposed. Extremely low tree
density does not match with any of the LUTs (Table 2) and the predicted understory signal does not match well with *in situ*
measurements. *In situ* measurements at Montiers were carried during leaf-off part of the season, which allowed full exposure
of the understory. Despite this, the predicted understory NDVI range from the MODIS data did not overlap with the *in situ*
measurements at Montiers at all. However, the MODIS BRDF values for these sites were marked with lower data quality flags
(QA>1), which correctly signals a decrease in accuracy in the calculations of the understory reflectance as well. Overall, our
results confirm that under conditions of very low tree density/leaves-off conditions, the understory signal can be assumed
identical with the total scene NDVI.

The performance of the method turns out to be limited over sites with a closed canopy such as Bílý Kříž (CZ-BK1), Hesse
(FR-Hes), or Vielsalm (BE-Vie) (Figure 5). This is because shadowing effect is the dominant scattering mechanism in such
stands, and understory carries only a negligible influence on the top-of-canopy signal. Bosco Fontana (IT-BFt) is another
broadleaf forest site with very high foliage cover (FC=0.91), yet the predicted understory NDVI range entirely overlaps with





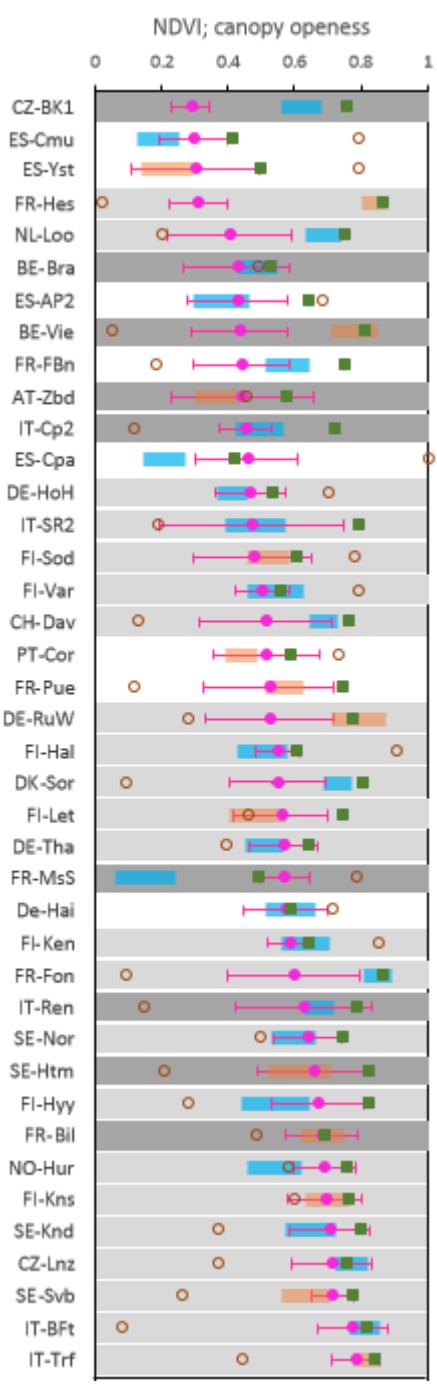

**Figure 5.** Understory NDVI (NDVIu) ranges (see Section 2.2 how the ranges are obtained; blue bars for site representative retrievals; orange bars for possible non site representative retrievals), *in situ* measurements (purple dots), computed nadir NDVI values from MODIS BRDF/albedo data (green squares). Canopy openness (1-fraction of foliage cover) shown with brown open circles. MODIS BRDF parameters with lower quality flags are indicated with light gray (QA=1) or dark gray (QA>1) bars. QA = 0 – best quality, full inversion; QA = 1 good quality, full inversion (including the cases with no clear sky observations over the day of interest and those with a Solar Zenith Angle > 70°); QA > 1 – lower quality magnitude inversion using archetypal BRDF shapes (for details please see Schaaf et al. (2002).





the collected *in situ* values. It should be noted that in contrast to other sites with closed canopies, Bosco Fontana has a very
dense vegetation throughout the full vertical profile, and no clear distinction between overstory and understory can be made.
Our results (Figure 5) indicate that in general reliable, independent retrieval of understory signal is not possible if foliage cover
exceeds 85 %.

MODIS BRDF data were of the best quality in the case of Font Blanche site (FR-FBn). The canopy here was also relatively
open (FC=0.18), yet the predicted understory NDVI range was higher than the lower NDVI captured by the *in situ* understory
measurements. The Font Blanche site has a dense intermediate layer dominated by juvenile holm oaks (*Quercus ilex L.*) with
a mean height of 6 m (Figure 7). Although the site was deemed spatially homogeneous at MODIS footprint scale (Table 2),
the tall, dense layer made it impossible to obtain truly representative *in situ* measurements of understory reflectance. A similar
situation with a tall shrub layer and thus a mismatch between the available *in situ* measurements and the predicted range of
understory NDVI values was also encountered at another pine-dominated, spatially homogeneous site at Loobos (NL-Loo).
Under such conditions the understory NDVI values retrieved with EO data might actually provide a more complete picture of
understory condition.

MODIS BRDF data were also of the best quality over the Coruche (PT-Cor) site with a very open canopy (FC= 0.272). Such
scenario should be optimal for the understory signal retrieval; yet the *in situ* measured NDVI of the understory is still higher
than the predicted range with MODIS BRDF data. This disagreement appears to be caused by the presence of a water reservoir
within the footprint of the MODIS pixel overlapping this site, which has contributed to lower reflectance in the red and NIR
part of the spectrum. The water surface was not sampled during *in situ* measurements. Similar effect of nearby lake can be also
observed in case of the Hurdal (NO-Hur) site.

As discussed in section 2.5, clear-felling was carried near the Wüstebach (DE-RuW) tower in 2013. Our assessment of site
homogeneity showed that the site cannot be considered spatially homogeneous at the gridded 500 m spatial resolution of
MODIS pixels (Figure 3D). The clear-felling action exposed the understory and encouraged the growth, resulting in an overlap
of the total and retrieved understory signal by MODIS BRDF data. *In situ* measurements carried within the still forested part
of the site around the tower with greater canopy closure resulted in lower understory NDVI values. The Wüstebach site
illustrates the importance of taking into account the spatial heterogeneity of a given site while comparing *in situ* measurements
with EO observations at the corresponding scale. The proposed framework by Román et al. (2009) and Wang et al. (2017)
using semivariograms is an efficient tool for evaluating site spatial representativeness.

In summary, while the understory retrieval algorithm was originally developed for conditions within the boreal region forests
(Canisius and Chen, 2007), Figure 5 suggests encouraging performance of the retrieval algorithm over a much wider range of





different forest sites. Reliable retrievals of forest understory appear to be feasible while taking into account the limitations due to site heterogeneity, foliage cover, and input data quality.

**3.3 Seasonal courses**

Figure 6 offers the overview of seasonal dynamics of understory for six select sites over the full latitudinal range (67°N - 38°N) across Europe.

The Station for Measuring Ecosystem-Atmosphere Relations in Värriö (Fi-Var) located in northern Finland (67°46'N, 29°35'E), represents a subarctic climate regime near the northern timberline. This site experiences very rapid increases in NDVI values at the beginning of the growing season (Figure 6A). This is linked with the disappearance of snow and exposure of the underlying understory vegetation, consisting predominantly of moss and lichen. The overstory coverage by Scots pine trees is sparse, and the overall NDVI signal fluctuations during the year are governed by understory layer. This site is also often covered with clouds, which prevents acquisition of large number of good quality MODIS observations. However, *in situ* understory NDVI measurements fall within the predicted NDVI range when the MODIS data were acquired early in the growing season (DOY 165), even despite a lower quality of MODIS data (QA=1). No MODIS observations can be made of this site after DOY 259, due to insufficient amount of light.

The forest floor is mainly covered by moss at Norunda (SE-Nor) as well. While understory NDVI values reach similar values in summer at both sites (Figure 6A-B), the snow disappears earlier at Norunda which results in earlier onset of higher understory NDVI values. *In situ* measurements fit very well within the predicted range with MODIS data on DOY 295. While Värriö has a higher tree density (748 trees/ha) than Norunda (600 trees/ha), Norunda, dominated with Norway spruce trees, has much higher foliage cover (FC=0.5 at Norunda compared to FC=0.21 at Värriö). The understory signal contribution is smaller and the total NDVI is higher at Norunda. The Norunda site provides yet another excellent demonstration of the influence of MODIS data quality on the understory NDVI retrievals as well. There was a repeated unrealistic fluctuation of understory NDVI values with the period of DOY 229-260 (Figure 6B). The MODIS BRDF data during that period were marked with lower data quality flags (QA>1).

Very good agreement is observed among *in situ* measurements, understory and total NDVI signal from MODIS (Figure 6C) at Hainich (DE-Hai) at the beginning of the growing season (DOY=102). The site is dominated with deciduous beech trees, which were leafless when the *in situ* measurements were taken. This allowed a full exposure of the understory which dominated the total reflectance signal from the stand during that moment in the season. The understory coverage was a mixture of litter and sprouting green understory. Later on, Hainich test site develops quickly an overstory layer with LAI values reaching up to 5 (Pinty et al., 2011). Such dense layer would prevent the retrieval of true understory signal with our methodology. This is



**Figure 6.** Seasonal courses of understory NDVI (NDVIu) ranges (blue bars for site representative retrievals; orange bars for possible site non-representative retrievals), nadir NDVI values from MODIS BRDF/albedo data (green lines), *in situ* measurements (purple dots) over select study sites. Gray bars mark MODIS BRDF parameters with lower quality flags (light gray, QA=1; dark gray, QA>1); black bars - no data available.



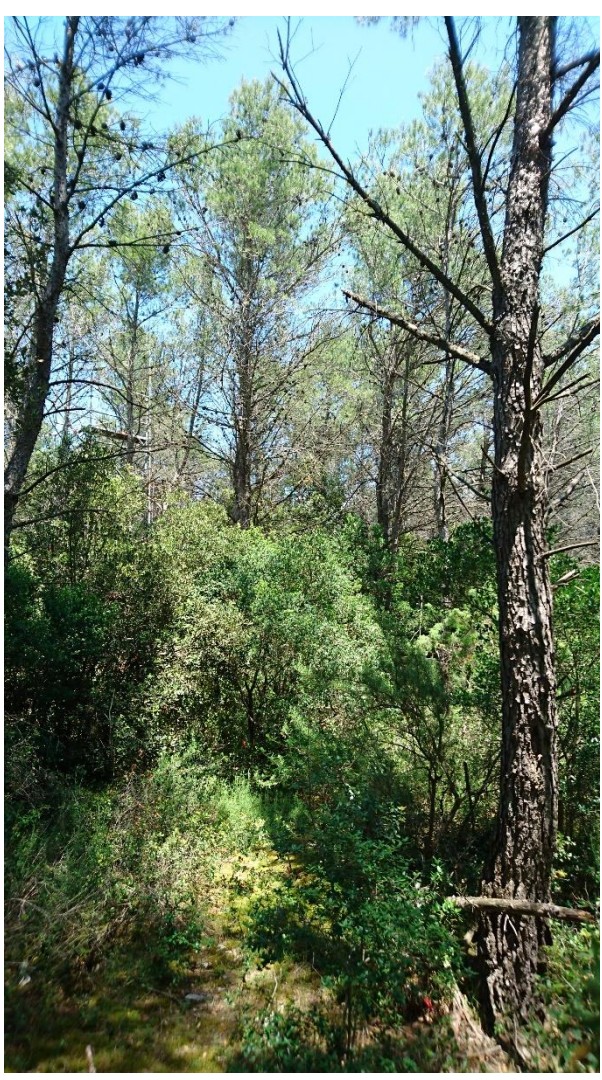

**Figure 7.** Foliage stratification between overstory and understory bushes at ICOS site Font Blanche in south-east France. Please note that the understory *in situ* measurements reported in Figure 6 did not include the tall (h>2 m) bushes located under the *Pinus halepensis* dominated overstory.

confirmed by the unrealistic very close agreement between the very high NDVI values (NDVI ~ 0.9) obtained from the total and understory signal at Hainich test site during the peak of growing season (Figure 6C). However, this shortcoming would be mitigated by the fact that in such cases the understory may be negligible in terms of LAI and overall contribution to the total signal.

Clear difference can be observed between the total and understory NDVI values at Font Blanche (FR-FBn) for most of the growing season (Figure 6D). The site is composed of an overstory top stratum (13 m) dominated by Aleppo pines (*Pinus*



*halepensis Mill*), an intermediate stratum (6 m) dominated by holm oaks (*Quercus ilex L.*) and a shrub stratum (Simioni et al., 2020). Rainfall occurs mainly during autumn and winter with about 75% between September and April. The higher NDVIu values derived from the MODIS BRDF data during the period DOY 120-160 may be treated with caution due to the more frequent flagging of the MODIS inputs with lower data quality flags (Figure 6D). Even lower understory NDVI values (NDVIu

range 0.15-0.35) occur at another site with a Mediterranean-type climate, Yeste in Spain (Figure 6E). The seasonal course of NDVI values here, with a low variation, is quite similar with Font Blanche. The increase in understory NDVI values in the autumn from DOY 288 is linked with the on-set of the rainfall period.

As illustrated above, given the high quality of MODIS BRDF data, the understory signal retrieval method performs well with

forests with open canopy. However, it is not quite possible to separate understory signal in closed canopies. There is an obvious disagreement between available *in situ* measurements and the predicted understory NDVI range at Hesse (FR-Hes) (Figure 6F) which could be only partly explained by the insufficient spatial homogeneity of the site at the MODIS pixel footprint (Table 2). Hesse has a high foliage cover (FC = 0.98), LAI up to 7 and tall trees (h > 23 m). Understory would then have a negligible influence on the top-of-canopy signal. The visibility and contribution of understory signal also diminishes even

further at off-nadir viewing directions (Rautiainen et al., 2008). Figure 6F confirms that in such situations the retrieval method cannot provide the correct, independent estimation of the understory signal. At the same time it should be noted that for closed canopies the understory signal (or lack of information about it) is not critical for the retrieval of biophysical properties of prime interest—LAI and fAPAR of the upper forest canopy layer with remote sensing (Garrigues et al., 2008; Weiss et al., 2014).

## 4 Conclusions

We report on the performance of a physically based approach to estimate understory NDVI from daily MODIS BRDF/albedo data, at a 500 m gridded spatial resolution, over the extended network of the Integrated Carbon Observing System (ICOS) forest ecosystem sites, distributed along wide latitudinal and elevational ranges (68°N - 38°N, 12-1864 m a.s.l.) across Europe. The analysis corresponds to a Stage 1 validation as defined by the CEOS (Nightingale et al., 2011; Weiss et al., 2014). The method can deliver good retrievals especially over different forest types with open canopies. The performance of the method

was found limited over forests with closed canopies (high foliage cover), where the signal from understory gets much attenuated. Our results illustrate the importance of considering both the spatial heterogeneity of the field site, as well as limitations and documented quality of the MODIS BRDF product. The results from the *in situ* measurements of understory layer can be valuable, in themselves, as source of information over the wide array of forest understory conditions contained within the tower footprints of individual ICOS forest ecosystem sites and serve as an input for improved modelling of local

carbon and energy fluxes.



*Data Availability.* Data will be made available online in the SPECCHIO information system (https://specchio.ch/) upon publication (currently under review).

*Author contributions.* JP conceived the project, collected data, ran data analysis and interpretation, and led the writing of manuscript. AE carried the spatial representativeness analysis. LK analysed the forest canopy cover/closure analysis. NH helped with the field collection at Hohes Holz. All co-authors discussed the results and contributed to writing the manuscript. Authors after LK are listed in alphabetic order.

*Competing interests.* The authors declare that they have no conflict of interest.

*Acknowledgments.* This study was supported from Estonian Research Council Grant PUT1355 and Mobilitas Pluss MOBERC11. This research (field campaigns at Brasschaat, Kindla, Zoebelboden and Machuqueira do Grou/LTsER Montado platform) was co-funded by the Transnational Access scheme of eLTER (Horizon 2020 project grant agreement no. 654359). We acknowledge ICOS Sweden, co-funded by the Swedish Research Council (SRC) under the Grant number 2015-06020, for provisioning of measurement facilities and experimental support. The MODIS BRDF data are supported by NASA grant 80NSSC18K0642.

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
