# Peer review of "Retrieval and validation of forest background reflectivity from daily MODIS bidirectional reflectance distribution function (BRDF) data across European forests"

_Biogeosciences, 2020_

## Referee Comment (RC1) · Anonymous Referee #1 · 30 Dec 2020

Pisek et al. provides the description and validation for their method for retrieving under-story NDVI values for a variety of forest types and seasons from MODIS BRDF data. They found that their method produces good estimates for open canopies, but that it had limited performance for closed canopies and that MODIS data quality and spatial heterogeneity impact the performance.

Overall, I found the study and manuscript straight-forward, concise, and with a high potential to be impactful. Thus, I do not have any major comments and most of the comments I have relate to improving the readability of the manuscript. There were
several instances where making the language more consistent would help. Specifically, I would choose to refer to the canopy by the openness or the foliage coverage instead of switching between the two, which can be confusing or at least requires a little extra thought. Figure 5 shows the data in terms of openness, but the main text often refers to it as foliage coverage. This became an issue in the sentence on line 300-301 where I believe there is a mismatch between this assertion that FR-FBn has high openness (and low FC) and the low openness value that is shown in Figure 5.

Specific comments:

Line 155: the ending of the sentence was cut off

Line 211: missing spaces between values and "nm" units

Line 242: missing space between 0.5 and "km" unit

Figure 3: Font needs to be bigger especially in panels b and d.

Figure 5: It took me a bit to understand what many of the individual parts of the figure indicated and I do not think it is self-explanatory enough as a figure and caption. I found most of the caption initially confusing, specifically the concept of "understory NDVI ranges." (1) I think it should be explicitly stated that the understory NDVI ranges are what you predicted/estimated and that these are for selected days (the ones you measured). I would consider changing the caption to be closer to "estimated understory NDVI ranges for selected days are given in blue bars for sites that were well represented and orange bars for sites that were poorly represented..." (2) I do not think that green squares should be used for the computed nadir NDVI values because this makes it look like they are intervals (like the "ranges") when they only represent point values. They have no uncertainty associated with them based on my understanding and, thus, another shape should be selected. Additionally, it was initially not clear to me what these values were and I think it should be clarified in the caption that these are the total NDVI values, which is how you refer to them in the text. (3) You reference

section 2.2 for how the ranges were obtained, but I think you meant to reference 2.3 as 2.2 does not include a description of the estimated ranges (only the in situ measured ones). (4) Further description of how the in situ measurements are displayed is required, specifically what the point (mean or median?) and bars (I think the +/- 1 standard deviation based on the text) indicate. (5) I would consider reordering the rows/sites. Currently it is ordered by increasing in situ measurements, but the text does not include a discussion around this. Instead I would consider ordering based on the canopy openness because this would help illustrate your result in line 297-298 that the retrieval of understory signal is not accurate if foliage cover exceeds 85%. This is only a suggestion though.

Figure 6: I have the same comment has in Figure 5 around making the labels in the caption more explicit.

―――――――――――――――――――――――

---

## Referee Comment (RC2) · Alexei Lyapustin (Referee) · 2 Jan 2021

[referee-annotated manuscript omitted]

---

## Author Comment (AC1) · 8 Jan 2021

Below we provide our replies to the requested edits. We thank the reviewer for his/her constructive comments that help to improve our original submission. The instructions for our final response state that revised manuscript should not be prepared/provided at this stage. For clarity, in our replies we provide the intended revised versions of the individual sentences following the reviewer's suggestions.

Responses to Reviewer #1 Pisek et al. provides the description and validation for their

method for retrieving understory NDVI values for a variety of forest types and seasons from MODIS BRDF data. They found that their method produces good estimates for open canopies, but that it had limited performance for closed canopies and that MODIS data quality and spatial heterogeneity impact the performance. Overall, I found the study and manuscript straight-forward, concise, and with a high potential to be impactful. Thus, I do not have any major comments and most of the several instances where making the language more consistent would help. Specifically, I would choose to refer to the canopy by the openness or the foliage coverage instead of switching between the two, which can be confusing or at least requires a little extra thought. Figure 5 shows the data in terms of openness, but the main text often refers to it as foliage coverage. This became an issue in the sentence on line 300-301 where I believe there is a mismatch between this assertion that FR-FBn has high openness (and low FC) and the low openness value that is shown in Figure 5.

REPLY: Accepted. We stick with the foliage coverage term throughout the whole text now. Overstory foliage cover will be plotted in Figure 5 instead of canopy openness.

Specific comments: Line 155: the ending of the sentence was cut off

REPLY: Accepted. We apologize for the omission. Here is the full sentence with the missing part added: When a tuning parameter (called "structuring element" in image processing) was set so that "large" gaps only occurred between individual tree crowns (Korhonen and Heikkinen, 2009), the proportions of gaps inside and between individual crowns could be calculated.

Line 211: missing spaces between values and "nm" units

REPLY: Accepted. The missing spaces will be added to the revised manuscript.

Line 242: missing space between 0.5 and "km" unit

REPLY: Accepted. The missing spaces will be added to the revised manuscript.

Figure 3: Font needs to be bigger especially in panels b and d.

Reply: Accepted. We increase the figures and fonts to correspond to the layout and design to match the one in the original work by Roman et al. (2009, RSE; Fig, 7). Please see we have a limited budget to cover the APCs and so have to try to make our manuscript as compact as possible. Thank you for understanding.

Figure 5: It took me a bit to understand what many of the individual parts of the figure indicated and I do not think it is self-explanatory enough as a figure and caption. I found most of the caption initially confusing, specifically the concept of "understory NDVI ranges." (1) I think it should be explicitly stated that the understory NDVI ranges are what you predicted/estimated and that these are for selected days (the ones you measured). I would consider changing the caption to be closer to "estimated understory NDVI ranges for selected days are given in blue bars for sites that were well represented and orange bars for sites that were poorly represented. . ."

REPLY Accepted. We change the caption to state this explicitly as suggested by the reviewer: 'Estimated understory NDVI (NDVIu) ranges for selected days (see Section 2. 3 how the ranges are obtained; blue bars for site representative retrievals; orange bars for possible non site representative retrievals).

(2) I do not think that green squares should be used for the computed nadir NDVI values because this makes it look like they are intervals (like the "ranges") when they only represent point values. They have no uncertainty associated with them based on my understanding and, thus, another shape should be selected. Additionally, it was initially not clear to me what these values were and I think it should be clarified in the caption that these are the total NDVI values, which is how you refer to them in the text.

Reply: Accepted. Good point. We change the symbol (cross instead of square) and expand the description in the caption as well: computed nadir total (understory+overstory) NDVI values from MODIS BRDF/albedo data (green crosses).

(3) You reference section 2.2 for how the ranges were obtained, but I think you meant to reference 2.3 as 2.2 does not include a description of the estimated ranges (only the

in situ measured ones).

REPLY: Accepted. Yes, the reviewer is correct: Section 2.3 should have been referred. We correct it.

(4) Further description of how the in situ measurements are displayed is required, specifically what the point (mean or median?) and bars (I think the +/- 1 standard deviation based on the text) indicate.

REPLY: Accepted. We provide the further description in the caption: in situ measurements (mean +/- 1 standard deviation shown in purple).

(5) I would consider reordering the rows/sites. Currently it is ordered by increasing in situ measurements, but the text does not include a discussion around this. Instead I would consider ordering based on the canopy openness because this would help illustrate your result in line 297-298 that the retrieval of understory signal is not accurate if foliage cover exceeds 85%. This is only a suggestion though.

REPLY: Accepted. Following the reviewer's suggestion, we reorder the sites by decreasing overstory foliage cover.

Figure 6: I have the same comment has in Figure 5 around making the labels in the caption more explicit.

REPLY: Accepted. We agree to make the caption description more explicit.

---

## Author Comment (AC2) · 8 Jan 2021

We thank the reviewer Alexei Lyapustin for his comments and suggestions that would help to improve our original submission. The instructions for our final response state that revised manuscript should not be prepared/provided at this stage. For clarity, in our replies we provide the revised versions of the individual sentences following the reviewer's suggestions.

The paper by Pisek et al evaluates possibility of assessing the understory NDVI using

site-level ground characterization and MODIS BRDF data (MCD43). Overall, it's a large work, the results are reasonable and deserve publication. My main comments are following (details are provided in the file attached): 1) Please explain the method in more detail. For instance, I've got an impression that the reflectances of understory and trees in the retrieval model are assumed Lambertian. If that's true then it should be explained, as well as the limitations of such assumption.

REPLY: We do not claim the vegetation targets (overstory, understory) are Lambertian reflectors (reflecting electromagnetic radiation equally in ALL DIRECTIONS), because they are not. At the same time, the retrieval approach relies on an assumption that at the given illumination geometry, there can exist viewing directions where reflectance factors would differ little between selected viewing angles. Several previous studies (e.g. Bacour & Bréon, 2005; Deering et al., 1999; Peltoniemi et al., 2005) found forward-scattering reflectance factors of various targets off the principal plane to be fairly constant. The most suitable viewing configuration for the understory signal retrieval (and the one used in this study as well) has been identified by Pisek et al. (2015, RSE) using a high angular resolution BRF dataset of Kuusk et al. (2014) and accompanying in situ measurements of understory reflectance factors (Kuusk et al., 2013). Please note that in that respect there is nothing new about the applied methodology and the assumptions that come along with it. We made sure we include all the important references related to the methodology and assumptions (e.g. Canisius and Chen, 2007, RSE; Pisek et al., 2015, RSE). Following the reviewer's request, we will include additional text and references in the revised version.

2) Presently, you are just saying that the method works well for open canopies. Since certain statistics is accumulated, please provide an assessment of the accuracy for derived NDVI of understory. More importantly, provide the same the for the Red and NIR and reflectances which is much more valuable as the NDVI is a non-linear function.

REPLY: We understand the reviewer's point. Accuracy of our approach would depend on the information about the forest stands that serves as an input to retrieve the proportions of individual components (sunlit/shaded trees/understory). The better we know these parameters, the more likely we can provide accurate retrievals. Since the information such as tree density, height or tree crown parameters is not always known, we followed Gemmell's (2000, RSE) logic and opted to report a range/window of understory NDVI (NDVIu) values obtained with the combinations of parameter values from Table 2 for each site and date. Specifying the correct constraints (window) for background alone has been previously found to greatly reduce the errors in the estimation of overstory parameters (Gemmell, 2000, RSE). In that sense we consider our product sufficiently accurate at this stage if the in situ measured values are located within the range (blue, orange bars in our Figures 4 and 5) of values predicted using the MODIS BRDF data. In our Results/Discussion we focused on highlighting and explaining the cases when the in situ measured values were found outside these ranges, and we were able to track and identify sources of the disagreement (e.g. closed canopies, site non-representativeness, MODIS data marked with lower quality flags, etc.). We were trying to deliver what Gemmell (2000, RSE) had asked for – the constraints/window for background values alone. Please note that in our manuscript we focused and were able to explain the cases when the in situ measured values were found OUTSIDE the window of values obtained with the MODIS BRDF product. Yes, we can do what the reviewer is asking for and assess the accuracy using e.g. the mean of predicted understory NDVI values and compare it with in situ values. We were just not sure if such assessment would provide a true picture in this case. Also, please note that once we omit the closed canopy (FC > 0.85) or spatially non-representative sites, we end up with only around 20 sites. Filtering the sites by quality flags of MODIS BRDF product would reduce the number of sites even more. We were not convinced such a small pool offers a representative sample for such assessment. Also please note that at the same time (at least to our knowledge), this study offers the largest pool of sites with in situ measured reflectances reported in the literature so far. We would like to comply with the reviewer's wishes but we also want to make sure we do not provide any potentially misleading information just for sake of calculating/providing numbers, and that's

why we did not include them in our submission. Our limited assessment may still meet Validation Stage 1 as defined by the CEOS LPV subgroup.

Also, for paper to be of any value, please provide an assessment of threshold for the canopy fraction below which the method you think should work to the specified accuracy.

REPLY: Accepted. We provide the following statement with foliage cover value explicitly stated: 'The method can deliver reasonable retrievals over different forest types with canopies where foliage cover does not exceed 85 %.'

Alexei. Please also note the supplement to this comment: https://bg.copernicus.org/preprints/bg-2020-360/bg-2020-360-RC2-supplement.pdf

REPLY: the comments from the supplement are inserted and answered below.

L91 - Please, refer to Tucker

Reply: Accepted. We add reference to Tucker (1979).

L155 - the sentence is not finished.

REPLY: Accepted. Here is the full sentence with the missing part added: When a tuning parameter (called "structuring element" in image processing) was set so that "large" gaps only occurred between individual tree crowns (Korhonen and Heikkinen, 2009), the proportions of gaps inside and between individual crowns could be calculated.

L161 - Your subscripts are not intuitive - please explain them. I figured that T is for trees, and G is for ground. What Z stands for?

REPLY: Accepted. We re-arrange the sentence to make it clearer: which includes reflectivities of the sunlit crowns ($R\_T$), sunlit understory ($R\_G$), shaded crowns ($R\_{ZT}$,), and shaded understory ($R\_{ZG}$).

L175 - Are you assuming that Rt, Rg, Rzt and Rzg do not depend on view angles?

[Figure]

REPLY: In this retrieval approach we work with the assumption that there exists a viewing configurations where the reflectance factors vary little between SELECTED angles (not all angles, but the selected ones). We stated this on L472-474 in our original submission.

L210 - The kernels are Ross and Li - please refer to those.

REPLY: Accepted. We refer to the kernels as Ross and Li in the revised text.

L210 - What you mean is you computed the BRF at the top of canopy using MCD43A1. And then derived the understory reflectance using the formulas described above. Please say so.

REPLY: Accepted. The revised statement is as follows: We computed the bidirectional reflectance factor (BRF) at the top of canopy with the isotropic parameter and two (volumetric and geometric) kernel functions (Roujean et al., 1992) for MODIS band 1 (red, 620–670 nm) and band 2 (NIR, 841–876 nm). We used the Ross and Li kernels to reconstruct the bidirectional reflectance factor (BRF) values for required geometries (see section 2.3) for each date, and then derived the understory signal using the formulas described in Section 2.3.

L276 - What is LUTs? Table 2 has no LUTs.

REPLY: Accepted. The reviewer is right. The statement has been revised: 'Extremely low tree density does not match with any original generalized input parameter values in Table 2 and the predicted understory signal does not match well with in situ measurements.'

L280 - To what accuracy? I would prefer to see the separate accuracy assessment for Red and NIR bands. Reply: To assess the decrease in accuracy quantitatively, we would have needed the high quality (QA=0) MODIS inputs for the same day - when the in situ retrievals were collected. But these are not available.

L285 - "shadowing" cannot be a "scattering mechanism".

REPLY: Accepted. We revise the sentence: 'This is because shadowing effect makes diffuse scattering the dominant mechanism in such stands, and understory carries only a negligible influence on the top-of-canopy signal'.

L365 - Terrible explanation. You only explain green color - this is NDVI from NBAR. What is blue and orange? Is it NDVI computed from MODIS BRDF or something else? If yes, why Green line is not in the middle of, say orange, for Yeste?

REPLY: Accepted. We modifiy the figure caption following the suggestions by the Reviewer 1: Figure 6. Seasonal courses of estimated understory NDVI (NDVIu) ranges (blue bars for site representative retrievals; orange bars for possible site non-representative retrievals), nadir total (understory+overstory) NDVI values from MODIS BRDF/albedo data (green lines), in situ measurements (mean +/- 1 standard deviation shown in purple) over select study sites. Gray bars mark MODIS BRDF parameters with lower quality flags (light gray, QA=1; dark gray, QA>1); black bars - no data available.

The green line is not in the middle of the NDVIu range (in orange) for Yeste, because it marks the total NDVI signal (including overstory as well).

L403 - Please, provide CF where you think the method works reasonably well. Also, please specify to what accuracy (for NDVI, and Red and NIR reflectance).

REPLY: Accepted. We provide the following statement with foliage cover value explicitly stated: 'The method can deliver reasonable retrievals over different forest types with canopies where foliage cover does not exceed 85 %.'